# Deciphering the genetic interactions between Pou4f3, Gfi1, and Rbm24 in maintaining mouse cochlear hair cell survival

**Guangqin Wang[1,2], Yunpeng Gu[1,2], Zhiyong Liu[1,2,3]***

[1]Institute of Neuroscience, State Key Laboratory of Neuroscience, CAS Center for Excellence in Brain Science and Intelligence Technology, Chinese Academy of Sciences, Shanghai, China; [2]University of Chinese Academy of Sciences, Beijing, China; [3]Shanghai Center for Brain Science and Brain-Inspired Intelligence Technology, Shanghai, China

**\*For correspondence:**
Zhiyongliu@ion.ac.cn

**Competing interest:** The authors declare that no competing interests exist.

**Abstract** Mammals harbor a limited number of sound-receptor hair cells (HCs) that cannot be regenerated after damage. Thus, investigating the underlying molecular mechanisms that maintain HC survival is crucial for preventing hearing impairment. Intriguingly, *Pou4f3*[-/-] or *Gfi1*[-/-] HCs form initially but then rapidly degenerate, whereas *Rbm24*[-/-] HCs degenerate considerably later. However, the transcriptional cascades involving Pou4f3, Gfi1, and Rbm24 remain undescribed. Here, we demonstrate that *Rbm24* expression is completely repressed in *Pou4f3*[-/-] HCs but unaltered in *Gfi1*[-/-] HCs, and further that the expression of both POU4F3 and GFI1 is intact in *Rbm24*[-/-] HCs. Moreover, by using in vivo mouse transgenic reporter assays, we identify three *Rbm24* enhancers to which POU4F3 binds. Lastly, through in vivo genetic testing of whether Rbm24 restoration alleviates the degeneration of *Pou4f3*[-/-] HCs, we show that ectopic Rbm24 alone cannot prevent *Pou4f3*[-/-] HCs from degenerating. Collectively, our findings provide new molecular and genetic insights into how HC survival is regulated.

## eLife assessment

In this **valuable** study, the authors explore regulatory cascades governing mammalian cochlear hair cell development and survival. They confirm previous studies that the transcription factors Pou4f3 and Gfi1 are necessary for hair cell survival and use **compelling** evidence to demonstrate that the RNA-binding protein gene RBM24 is regulated by Pou4f3, but not Gfi1. These findings will be of interest to those working on hearing loss and hold significance for viral gene delivery methods aiming to manipulate gene expression.

## Introduction

Housed in the mammalian cochlea, the auditory epithelium—also known as the organ of Corti—contains the mechanosensory receptors hair cells (HCs) that detect sound information (*Wu and Kelley, 2012*; *Qiu and Müller, 2022*). Two types of HCs exist, the inner HCs (IHCs) and outer HCs (OHCs), and these appear in, respectively, a single row and three rows (*Groves et al., 2013*). Both OHCs and IHCs are derived from cochlear progenitors expressing high levels of ATOH1, a master transcription factor (TF) in HC development (*Li et al., 2022*; *Driver et al., 2013*; *Matei et al., 2005*), and no HCs form in the *Atoh1*[-/-] cochlea, highlighting the essential role of ATOH1 in specifying the general HC fate in the

undifferentiated progenitors (*Bermingham et al., 1999*; *Luo et al., 2022*). Whereas OHCs uniquely express the motor protein PRESTIN (encoded by *Slc26a5*) and function as sound amplifiers (*Zheng et al., 2000*; *Liberman et al., 2002*), IHCs specifically express FGF8, vGLUT3 (encoded by *Slc17a8*), and OTOFERLIN (*Pan et al., 2023*; *Roux et al., 2006*; *Jacques et al., 2007*; *Ruel et al., 2008*; *Seal et al., 2008*) and act as the primary sensory cells that form synapses with distinct subtypes of type I spiral (auditory) neurons (*Petitpré et al., 2018*; *Shrestha et al., 2018*; *Sun et al., 2018*; *Zhang et al., 2023*). Moreover, INSM1 is necessary for stabilizing the OHC fate, and ~50% of *Insm1*$^{-/-}$ OHCs tend to transdifferentiate into IHC-like cells (*Wiwatpanit et al., 2018*; *Li et al., 2023a*); conversely, IKZF2 is not required in early OHC development but is essential for consolidating the OHC fate, and *Ikzf-2*$^{cello/cello}$ mutant OHCs are dysfunctional and misexpress IHC genes (*Chessum et al., 2018*). INSM1 is an epistatic but indirect positive regulator of *Ikzf2*, and restoration of IKZF2 expression can partially mitigate the phenotypes of *Insm1*$^{-/-}$ OHCs (*Li et al., 2023a*). In contrast to INSM1 and IKZF2, TBX2 is required in IHC fate specification, differentiation, and maintenance, and IHCs convert into OHCs in the absence of TBX2 (*Bi et al., 2022*; *García-Añoveros et al., 2022*; *Kaiser et al., 2022*; *Li et al., 2023b*).

The survival of cochlear HCs must be maintained after their production. POU4F3, a POU-domain family TF encoded by the hearing-loss gene *DFNA15*, is essential for HCs to survive (*Vahava et al., 1998*; *Masuda et al., 2011*; *Xiang et al., 1997*; *Zhu et al., 2020*; *Erkman et al., 1996*). However, unlike ATOH1, POU4F3 is not involved in the fate determination of HCs (*Xiang et al., 1998*). As a target gene regulated by ATOH1 (*Masuda et al., 2011*; *Yu et al., 2021*), *Pou4f3* is expressed in HCs between embryonic day 14.5 (E14.5) and E16 in a basal-to-apical gradient, and POU4F3 then activates the expression of *Gfi1*, which encodes a zinc-finger TF (*Wallis et al., 2003*; *Hertzano et al., 2004*); Gfi1 is undetectable in *Pou4f3*-deficient HCs (*Hertzano et al., 2004*), and HCs, particularly OHCs, degenerate in *Pou4f3* and *Gfi1* mutants by birth (*Hertzano et al., 2004*). Besides POU4F3 and Gfi1, the RNA-binding protein RBM24 is also specifically expressed in cochlear HCs (*Grifone et al., 2018*). We recently showed that RBM24 is indispensable for maintaining OHC survival, but the OHC death in *Rbm24* mutants occurs considerably later than in *Pou4f3* and *Gfi1* mutants; OHCs develop normally until birth but most OHCs undergo cell death by postnatal day 19 (P19) in *Rbm24*$^{-/-}$ mice (*Wang et al., 2021*). How *Rbm24* expression is regulated during cochlear HC development is unknown.

The similar but delayed phenotype of OHC death in *Rbm24* mutants relative to that in *Pou4f3*- and *Gfi1*-deficient mice prompted us to dissect the potential genetic interactions among them by using an in vivo genetic approach. Our results showed that the onset of RBM24 expression was completely repressed in *Pou4f3*$^{-/-}$ HCs, but that, unexpectedly, RBM24 expression was normal in *Gfi1*$^{-/-}$ HCs. Moreover, the expression of neither POU4F3 nor GFI1 was altered in *Rbm24*$^{-/-}$ HCs. Thus, POU4F3, but not GFI1, is required in *Rbm24* expression. Furthermore, we identified three *Rbm24* enhancers that were sufficient to drive specific EGFP reporter expression in HCs, and these enhancers are likely bound by POU4F3. Lastly, we found that restoration of RBM24 expression alone cannot alleviate the degeneration of *Pou4f3*$^{-/-}$ HCs, which indicates that the expression of additional POU4F3-targeted genes must be restored to enable *Pou4f3*$^{-/-}$ HCs to survive. Our study provides new insights into the genetic interactions among *Pou4f3*, *Gfi1*, and *Rbm24*, which hold potential applications in HC protection.

## Results

### RBM24 expression is completely repressed in *Pou4f3*$^{-/-}$ cochlear HCs

HC degeneration occurs in both *Pou4f3*$^{-/-}$ and *Rbm24*$^{-/-}$ mice, with the phenotype appearing earlier and being more severe in *Pou4f3*$^{-/-}$ mice (*Hertzano et al., 2004*; *Wang et al., 2021*). This led us to speculate that genetic interaction exists between *Pou4f3* and *Rbm24*. We reasoned that if POU4F3 is an upstream positive regulator of *Rbm24*, RBM24 expression would be downregulated in the absence of POU4F3. To rapidly test this possibility, we exploited our previously established CRISPR-stop approach (*Zhang et al., 2018*). CRISPR-stop allows early stop codons to be introduced without inducing DNA damage through Cas9, which can cause deleterious effects (*Kuscu et al., 2017*). More importantly, the CRISPR-stop approach can generate Founder 0 (F0) mice carrying homozygous or mosaic homozygous gene mutations, and the F0 mice are thus immediately ready for phenotypic analysis, considerably faster than in the case with traditional gene-targeting methods (*Wang et al., 2021*; *Zhang et al., 2018*). Co-injecting one sgRNA (sgRNA-1) against *Pou4f3* and base-editor components

into one-cell-stage mouse zygotes yielded F0 mice, whose tail DNA was subject to Sanger sequencing (*Figure 1—figure supplement 1A*). Relative to wild-type (WT) mice (*Figure 1—figure supplement 1B*), the F0 mice with homozygous premature emergence of the stop codon TAG were defined as *Pou4f3*$^{-/-}$ mice and were immediately ready for phenotypic analysis (*Figure 1—figure supplement 1C*).

Triple staining of POU4F3, RBM24, and INSM1 revealed that OHCs in WT mice at E16.5 were POU4F3+/RBM24+/INSM1+, whereas IHCs expressed POU4F3 and RBM24 but not INSM1 (*Figure 1—figure supplement 1D–D'''*). Conversely, POU4F3 expression was absent in *Pou4f3*$^{-/-}$ mice (*Figure 1—figure supplement 1E–E'''*), confirming that POU4F3 translation was blocked, and unlike in WT mice, RBM24 expression was undetectable in the *Pou4f3*$^{-/-}$ mice at E16.5 (*Figure 1—figure supplement 1E–E'''*). Notably, INSM1 expression appeared normal in *Pou4f3*$^{-/-}$ mice (arrows in *Figure 1—figure supplement 1E–E'''*). Moreover, the presence of INSM1+ OHCs in the *Pou4f3*$^{-/-}$ mice eliminated the possibility that the absence of RBM24 was due to a secondary effect of OHC death or delayed differentiation, and also agreed with previous reports that initial HC differentiation can occur without POU4F3 (*Xiang et al., 1998*; *Hertzano et al., 2004*). Thus, the rapid *Pou4f3* loss-of-function analysis by using CRISPR-stop supported our hypothesis that POU4F3 is required for turning on RBM24 expression.

To further validate the observation described above, we constructed germline-stable *Pou4f3* mutants in which the entire *Pou4f3* genomic region between sgRNA-2 and sgRNA-3 was deleted (*Figure 1A*). The three obtained genotypes, *Pou4f3*$^{+/+}$, *Pou4f3*$^{+/-}$, and *Pou4f3*$^{-/-}$, were readily distinguished using tail-DNA PCR (*Figure 1B*). Relative to WT mice (*Figure 1C*), *Pou4f3*$^{-/-}$ mice again completely lacked RBM24 expression at E16.5 (*Figure 1D*). Similar to INSM1, BCL11B is an OHC marker (*Wiwatpanit et al., 2018*), and triple staining of RBM24, POU4F3, and BCL11B revealed that as in WT HCs (*Figure 1E–E'''*), BCL11B was normally expressed in OHCs lacking both POU4F3 and RBM24 at E16.5 (*Figure 1F–F'''*). This result again confirmed that nascent OHCs were normally produced in the absence of POU4F3. Moreover, in contrast to WT mice at P1 (*Figure 1—figure supplement 2A–A'''*), the germline *Pou4f3*$^{-/-}$ mutants showed HC degeneration at P0 (*Figure 1—figure supplement 2B–B''*) and P1 (*Figure 1—figure supplement 2C–C''*). Normalizing the numbers of the remaining HCs in the *Pou4f3*$^{-/-}$ mice at P0 or P1 against those in WT mice at P1 revealed that in P0 *Pou4f3*$^{-/-}$ mice, 35.94% ± 4.52%, 38.16% ± 2.79%, and 63.8% ± 5.28% of the HCs survived in basal, middle, and apical cochlear turns, respectively, whereas in P1 *Pou4f3*$^{-/-}$ mice, 5.54% ± 3.11%, 15.79% ± 3.65%, and 65.64% ± 7.95% of the HCs survived. HC degeneration in the middle and basal turns, but not the apical turn, was significantly more severe at P1 than at P0 (*Figure 1—figure supplement 2D*). Lastly, unlike in WT cochleae in which all MYO7A+ HCs expressed RBM24 (*Figure 1—figure supplement 2E–E''*), in *Pou4f3*$^{-/-}$ cochleae, the remaining HCs lacked RBM24 expression (arrows in *Figure 1—figure supplement 2F–F''*). Collectively, our results obtained using two distinct *Pou4f3*-deficient models indicated that POU4F3 is indispensable for turning on RBM24 expression. However, whether GFI1 is also essential for RBM24 expression remained unknown.

## Construction of *Gfi1-3×HA-P2A-Cre/+* knockin mouse strain

We sought to determine whether POU4F3 regulates *Rbm24* expression through GFI1 or independently of GFI1. Because a suitable commercial GFI1 antibody for immunostaining was unavailable, we constructed a knockin mouse strain, *Gfi1*$^{3×HA-P2A-Cre/+}$ (*Gfi1*$^{HA-Cre/+}$ in brief), by using our routine CRISPR/Cas9 approach (*Figure 2—figure supplement 1A–C*); here, the GFI1 C-terminus was tagged with three hemagglutinin (HA) fragments and Cre expression was under the control of endogenous *cis*-regulatory elements (CREs) of *Gfi1*. The obtained WT (*Gfi1*$^{+/+}$), heterozygous (*Gfi1*$^{HA-Cre/+}$), and homozygous (*Gfi1*$^{HA-Cre/HA-Cre}$) mice were readily distinguished using tail-DNA PCR (*Figure 2—figure supplement 1D*). Southern blotting revealed that in addition to being inserted in the *Gfi1* locus, the targeting vector (*Figure 2—figure supplement 1B*) was randomly inserted in an unknown genomic region; however, the random insertion likely occurred in a silent genomic region as indicated by the analysis discussed below.

Dual staining of HA (GFI1) and the pan-HC marker MYO7A revealed that, similar to *Gfi1*$^{+/+}$ mice (*Figure 2—figure supplement 1E–E''*), *Gfi1*$^{HA-Cre/HA-Cre}$ mice showed normal HC development at P1 (*Figure 2—figure supplement 1F–F''*). Moreover, HA (GFI1) was highly expressed in all MYO7A+ IHCs and OHCs in *Gfi1*$^{HA-Cre/HA-Cre}$ mice (arrows in *Figure 2—figure supplement 1F–F''*). This finding

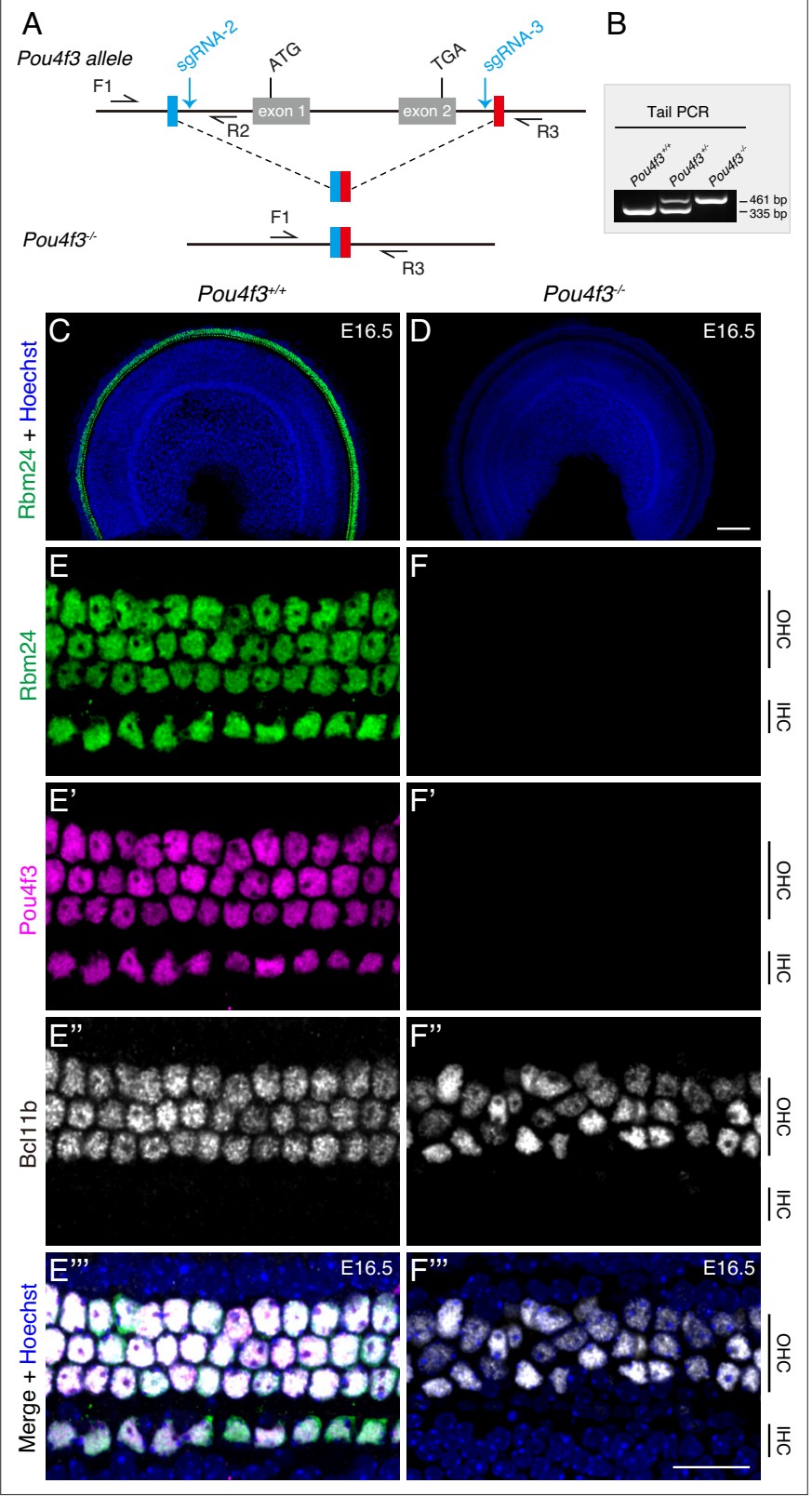

**Figure 1.** RBM24 expression is repressed in cochlear hair cells (HCs) in the absence of POU4F3. (**A**) Simple illustration of how *Pou4f3⁻/⁻* strain was generated. (**B**) One example gel image of tail-DNA PCR used to distinguish WT (*Pou4f3⁺/⁺*), heterozygous (*Pou4f3⁺/⁻*), and homozygous (*Pou4f3⁻/⁻*) mice. (**C–F‴**) Triple staining of RBM24, POU4F3, and BCL11B in WT (**C, E–E‴**) and *Pou4f3⁻/⁻* (**D, F–F‴**) mice (n = 3 for each) at E16.5; only the RBM24

*Figure 1 continued on next page*

*Figure 1 continued*

(green) channel (together with nuclear staining) is shown in (**C, D**) at low magnification. (**E–E′′′**) and (**F–F′′′**): High-magnification images of basal turn in WT and *Pou4f3*<sup>-/-</sup> mice, respectively. In the absence of POU4F3, RBM24 expression completely disappeared, although nascent BCL11B+ outer hair cells (OHCs) were present. IHC: inner hair cell. Scale bars: 100 μm (**D**), 20 μm (**F′′′**).

The online version of this article includes the following source data and figure supplement(s) for figure 1:

**Source data 1.** The original agarose gel image in *Figure 1B* (*Pou4f3*<sup>+/+</sup>, *Pou4f3*<sup>+/-</sup>, and *Pou4f3*<sup>-/-</sup>).

**Source data 2.** File containing *Figure 1B* and the original agarose gel analysis with highlighted bands and sample labels.

**Figure supplement 1.** RBM24 expression is repressed in cochlear hair cells (HCs) at E16.5 when POU4F3 is inactivated.

**Figure supplement 2.** Severe degeneration of *Pou4f3*<sup>-/-</sup> hair cells (HCs) occurs by perinatal ages.

**Figure supplement 2—source data 1.** File containing raw counts of the cochlear hair cells reported in the graph of *Figure 1—figure supplement 2D*.

---

confirmed that Gfi1 endogenous expression is unaffected in *Gfi1*<sup>HA-Cre/HA-Cre</sup> mice, which represents an advantage over the previous *Gfi1*<sup>Cre/+</sup> strain in which one copy of *Gfi1* is lost and early onset of hearing loss occurs (*Matern et al., 2017*; *Yang et al., 2010a*).

We also performed fate-mapping analysis in *Gfi1*<sup>HA-Cre/+</sup>; *Rosa26-loxp-stop-loxp-tdTomato* (Ai9)/+ (*Gfi1*<sup>HA-Cre/+</sup>; Ai9/+) mice at P2. The *Gfi1*<sup>HA-Cre/+</sup>; Ai9/+ model allowed us to both visualize the temporal GFI1 protein-expression pattern and permanently mark cells expressing *Gfi1* by using tdTomato. Neither tdTomato nor HA was detected in the MYO7A+ HCs of control Ai9/+ mice (*Figure 2—figure supplement 1G–G′′′ and I–I′′*), whereas the majority of MYO7A+ HCs expressed tdTomato and HA in the *Gfi1*<sup>HA-Cre/+</sup>; Ai9/+ mice (*Figure 2—figure supplement 1H–H′′′ and J–J′′*). HA (GFI1) was generally expressed more highly in IHCs than OHCs (*Figure 2—figure supplement 1J–J′′*), and the HA (GFI1) expression level exhibited manifest heterogeneity among the OHCs. We identified the OHCs expressing the highest (#1), intermediate (#2), and lowest (#3) levels of HA (GFI1) (arrows in *Figure 2—figure supplement 1J–J′′*), and we observed that, notably, such heterogenous GFI1 expression in neonatal HCs was not detected in *Gfi1*<sup>HA-Cre/HA-Cre</sup> mice (*Figure 2—figure supplement 1F–F′′*), likely due to the higher level of HA-tagged GFI1 in *Gfi1*<sup>HA-Cre/HA-Cre</sup> than *Gfi1*<sup>HA-Cre/+</sup> mice. Besides the tdTomato+ HCs, we detected tdTomato+ cells in nonsensory regions (arrows in *Figure 2—figure supplement 1H*), consistent with a previous report (*Matern et al., 2017*). Collectively, these results indicated that the *Gfi1*<sup>HA-Cre/+</sup> strain was suitable for visualizing GFI1 protein by using an anti-HA anti-body, although random insertion of the targeting vector also exists here. Alternatively, *Gfi1*<sup>HA-Cre/+</sup> can at least be treated as a pseudo-transgenic mouse strain that can be used to reliably visualize GFI1 and trace cells expressing *Gfi1*.

## GFI1 expression is prevented in *Pou4f3*<sup>-/-</sup> cochlear HCs

Because HA faithfully represented GFI1 expression in the *Gfi1*<sup>HA-Cre/+</sup> strain, we expected to observe repression of HA expression in *Pou4f3*<sup>-/-</sup> HCs, as reported previously (*Wallis et al., 2003*; *Hertzano et al., 2004*). We confirmed this by using our CRISPR-stop approach (*Zhang et al., 2018*). The experimental pipeline was mostly identical to that used for producing the *Pou4f3* mutants (*Figure 1—figure supplement 1*), except that the zygotes here were derived from male *Gfi1*<sup>HA-Cre/+</sup> mice (*Figure 2—figure supplement 2A*). We obtained mosaic (or chimeric) *Pou4f3*<sup>-/-</sup> mice, as confirmed through Sanger sequencing of tail-DNA PCR samples (*Figure 2—figure supplement 2B and C*), partly because the injection time was the late stage of the one-cell-stage zygotes. In control *Gfi1*<sup>HA-Cre/+</sup> mice, a pure 'C' base was present (blue arrow in *Figure 2—figure supplement 2B*), but a mixture of 'C' and 'T' double peaks existed in *Gfi1*<sup>HA-Cre/+</sup>; *Pou4f3*<sup>-/-</sup> (mosaic) mice (red arrow in *Figure 2—figure supplement 2C*).

In control *Gfi1*<sup>HA-Cre/+</sup> mice at E16.5, all HCs expressed POU4F3, RBM24, and HA (GFI1), although the HA (GFI1) levels again appeared heterogenous among the OHCs (*Figure 2—figure supplement 2D–D′′′*). By contrast, in the *Gfi1*<sup>HA-Cre/+</sup>; *Pou4f3*<sup>-/-</sup> (mosaic) mice, we detected HCs that had either lost or maintained POU4F3 expression (*Figure 2—figure supplement 2E–E′′′*). Notably, POU4F3+HCs expressed HA (GFI1) and RBM24 (blue arrows in *Figure 2—figure supplement 2E–E′′′*), whereas both HA (GFI1) and RBM24 were absent in HCs that had lost POU4F3 expression (orange arrows in

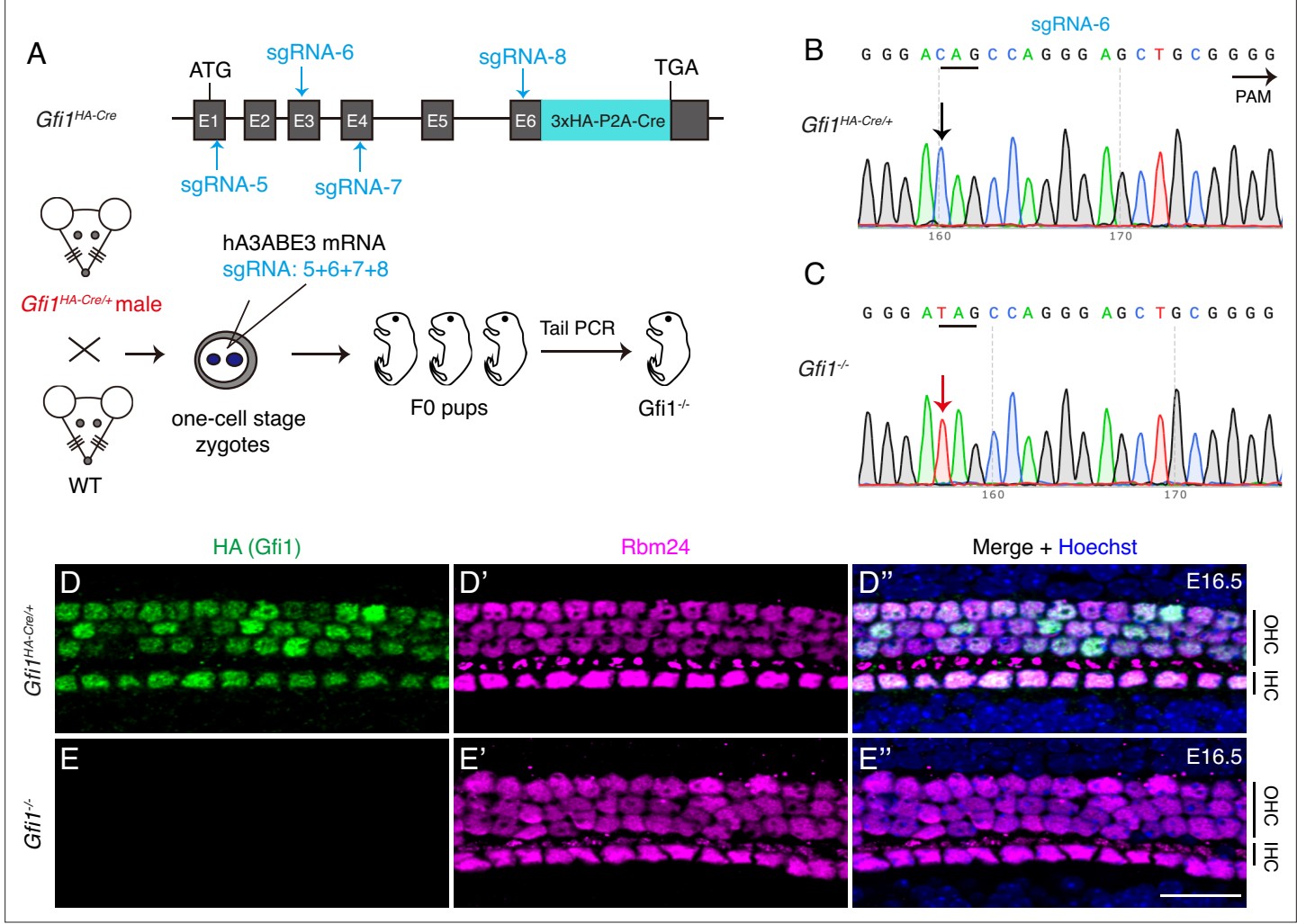

**Figure 2.** GFI1 is dispensable for RBM24 expression. (**A**) Simple cartoon depicting how *Gfi1* was inactivated in one-cell-stage zygotes derived from male *Gfi1^HA-Cre/+* mice by using the CRISPR-stop approach. (**B, C**) Using sgRNA-6 as an example, Sanger sequencing chromatograms of control *Gfi1^HA-Cre/+* (**B**) and F0 *Gfi1^-/-* (**C**) samples are presented. The base 'C' (black arrow in **B**) was converted to 'T' (red arrow in **C**), resulting in pre-emergence of the translation stop codon TAG. The red 'T' appears as a single peak, indicating that the stop codon pre-emerged in both alleles. (**D–E″**) Dual staining of HA (GFI1) and RBM24 in cochleae from control *Gfi1^HA-Cre/+* (**D–D″**, n = 3) and F0 *Gfi1^-/-* (**E–E″**) mice (n = 4) at E16.5. RBM24 expression (**E′**) was not altered in the absence of HA (GFI1) expression (**E**). OHC: outer hair cell; IHC: inner hair cell. Scale bar: 20 µm (**E″**).

The online version of this article includes the following source data and figure supplement(s) for figure 2:

**Figure supplement 1.** Construction of *Gfi1^3◊HA-P2A-Cre/+* (*Gfi1^HA-Cre/+*) mouse model in which HA expression faithfully represents GFI1 expression.

**Figure supplement 1—source data 1.** The original agarose gel image in *Figure 2—figure supplement 1D* (*Gfi1^+/+*, *Gfi1^HA-Cre/+*, and *Gfi1^HA-Cre/ HA-Cre*).

**Figure supplement 1—source data 2.** File containing *Figure 2—figure supplement 1D* and the original agarose gel analysis with highlighted bands and sample labels.

**Figure supplement 2.** GFI1 expression is lost in *Pou4f3^-/-* hair cells (HCs).

**Figure supplement 3.** RBM24 expression is normal in the absence of *Gfi1*.

**Figure supplement 3—source data 1.** The original agarose gel image in *Figure 2—figure supplement 3B* (*Gfi1^+/+*, *Gfi1^+/-*, and *Gfi1^-/-*).

**Figure supplement 3—source data 2.** File containing *Figure 2—figure supplement 3B* and the original agarose gel analysis with highlighted bands and sample labels.

*Figure 2—figure supplement 2E–E'''*). This again confirmed that HA is a reliable readout for GFI1 expression and that the expression is sensitive to the loss of POU4F3. Moreover, the results supported the view that POU4F3 regulates *Rbm24* in a cell-autonomous manner.

## GFI1 is dispensable for RBM24 expression

Next, we produced *Gfi1*^-/- mutants by using the same CRISPR-stop approach (*Zhang et al., 2018*). One-cell-stage zygotes derived from male *Gfi1*^HA-Cre/+ mice were injected with base-editor components and four different sgRNAs located in distinct exons of *Gfi1* (*Figure 2A*). The reason for the combined use of four sgRNAs is detailed in the 'Discussion' section. With sgRNA-6, for example, Sanger sequencing of tail DNA revealed that relative to control *Gfi1*^HA-Cre/+ mice (black arrow in *Figure 2B*), F0 mice showed premature emergence of the TAG stop codon, which resulted in homozygous Gfi1 inactivation (red arrow in *Figure 2C*). All HCs expressed HA (GFI1) and RBM24 in control *Gfi1*^HA-Cre/+ mice (*Figure 2D–D''*), and, notably, RBM24 expression was maintained in *Gfi1*^-/- HCs, which showed no HA (GFI1) expression at E16.5 (*Figure 2E–E''*). This finding suggested that RBM24 expression does not depend on GFI1.

The advantage of this *Gfi1*^-/- model is that it allows rapid and direct confirmation of the absence of GFI1 expression in F0 mice. However, complete loss of GFI1 expression in all HCs cannot be guaranteed here at single-cell resolution through HA staining because only one allele of *Gfi1* was *Gfi1*^HA-Cre. Thus, to eliminate the possibility that only the *HA*-tagged *Gfi1* allele was mutated, we generated germline-stable *Gfi1* mutants in which the majority of the *Gfi1* DNA fragment between sgRNA-4 and sgRNA-9 was deleted (*Figure 2—figure supplement 3A*). The WT (*Gfi1*^+/+), *Gfi1*^+/-, and *Gfi1*^-/- mice were readily identified using tail-DNA PCR (*Figure 2—figure supplement 3B*). In agreement with previous findings (*Matern et al., 2020*), dual staining of RBM24 and POU4F3 revealed that relative to WT mice (*Figure 2—figure supplement 3C–C''*), *Gfi1*^-/- mice (*Figure 2—figure supplement 3D–D''*) showed severe degeneration of HCs at P1, particularly OHCs; this validated the successful generation of the *Gfi1*^-/- mice. Notably, the surviving POU4F3+ HCs maintained the expression of RBM24 (orange arrows in *Figure 2—figure supplement 3D–D''*). By contrast, we observed no significant difference between WT and *Gfi1*^-/- mice at E16.5: both WT and *Gfi1*^-/- HCs expressed POU4F3 and RBM24, and both WT and *Gfi1*^-/- OHCs expressed BCL11B (*Figure 2—figure supplement 3E–F'''*). This suggests that the degeneration of *Gfi1*^-/- HCs does not begin by E16.5. The presence of POU4F3 in *Gfi1*^-/- HCs agreed with the previously mentioned notion that *Pou4f3* is epistatic to *Gfi1*. Collectively, the results of analyses of both *Gfi1* mutant models support the conclusion that GFI1 is dispensable for RBM24 expression.

## POU4F3 and GFI1 expressions are normal in *Rbm24*^-/- cochlear HCs

Considering that POU4F3 is upstream and a positive regulator of *Rbm24*, we predicted that POU4F3 expression would be unaffected in *Rbm24*^-/- HCs. To test this, we established an *Rbm24*^-/- mouse strain by injecting one *Rbm24* sgRNA (sgRNA-10) into one-cell-stage zygotes derived from male *Gfi1*^HA-Cre/+ mice (*Figure 3A*). We selected *Gfi1*^HA-Cre/+ zygotes in order to concurrently assess GFI1 and POU4F3 expression patterns in *Rbm24*^-/- HCs. Relative to control *Gfi1*^HA-Cre/+; *Rbm24*^+/+ mice (*Figure 3B*), *Gfi1*^HA-Cre/+; *Rbm24*^-/- (mosaic) mice (*Figure 3C*) showed premature emergence of the TAG stop codon. Notably, Sanger sequencing of tail DNA revealed that the *Rbm24* mutation was mosaic because mixed 'T' and 'C' peaks existed (red arrow in *Figure 3C*).

The mosaic inactivation of RBM24 was further confirmed through triple staining of RBM24, HA, and POU4F3 (*Figure 3D–E'''*). Relative to the expression in control *Gfi1*^HA-Cre/+; *Rbm24*^+/+ mice (*Figure 3D–D'''*), RBM24 expression disappeared in a fraction of cochlear HCs (orange arrows in *Figure 3E–E'''*) but remained normal in other HCs (blue arrows in *Figure 3E–E'''*) of the *Gfi1*^HA-Cre/+; *Rbm24*^-/- (mosaic) mice at E17. Notably, regardless of whether RBM24 expression was inactivated or not in the cochlear HCs, the expression patterns of both HA (GFI1) and POU4F3 remained intact. This suggests that *Pou4f3* is epistatic to *Rbm24* and that inactivation of the *Rbm24* gene does not affect POU4F3 expression. Moreover, our results suggested that *Gfi1* and *Rbm24* do not interact genetically: *Gfi1* inactivation did not affect *Rbm24* expression (*Figure 2*, *Figure 2—figure supplement 3*), and vice versa (*Figure 3E–E'''*).

## GATA3 is downregulated and IKZF2 is upregulated in *Rbm24*-deficient HCs

We next determined whether the expression patterns of two additional genes, *Gata3* and *Ikzf2*, which are involved in OHC development and survival (*Li et al., 2023a*; *Chessum et al., 2018*; *Bi et al.,*

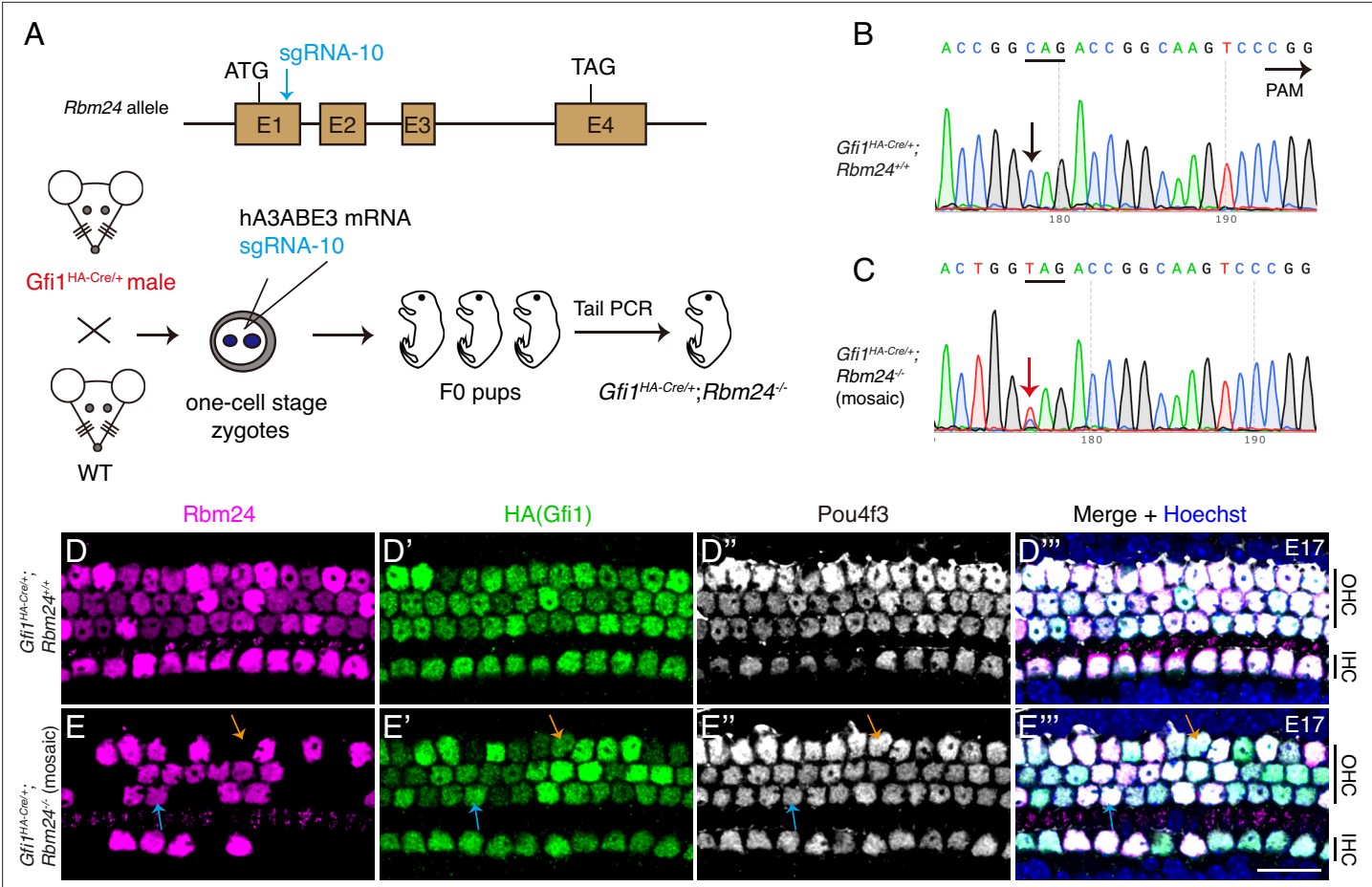

**Figure 3.** POU4F3 and GFI1 expressions are normal in the absence of *Rbm24*. (**A**) Simple cartoon showing how CRISPR-stop was used to induce mosaic inactivation of RBM24 in one-cell-stage mouse zygotes harboring *Gfi1^HA-Cre/+^*. Notably, the injection was in the late one-cell stage of the zygotes, which resulted in high rates of final mosaic inactivation. (**B, C**) Sanger sequencing chromatograms of control *Gfi1^HA-Cre/+^*; *Rbm24^+/+^* (**B**) and *Gfi1^HA-Cre/+^*; *Rbm24^-/-^* (mosaic) (**C**) samples. The base 'C' (black arrow in **B**) was converted to 'T' (red arrow in **C**) in a fraction of cochlear cells. Mosaic inactivation of *Rbm24* is evidenced by the double peaks of 'T' and 'C' (red arrow in **C**). (**D–E'''**) Triple staining of RBM24, HA, and POU4F3 in cochlear samples (basal turns) of control *Gfi1^HA-Cre/+^*; *Rbm24^+/+^* (**D–D'''**, n = 3) and *Gfi1^HA-Cre/+^*; *Rbm24^-/-^* (mosaic) (**E–E'''**, n = 3) mice at E17. Consistent with the Sanger sequencing result (**C**), RBM24 expression was found to be inactivated in cochlear hair cells (HCs) in a mosaic pattern (**E**). Orange arrows in (**E–E'''**): one outer hair cell (OHC) that lost RBM24 expression; blue arrows in (**E–E'''**): one OHC that maintained RBM24 expression. Regardless of whether RBM24 expression was inactivated or not, the expression of HA (GFI1) and POU4F3 was unaltered. IHC: inner hair cell. Scale bar: 20 µm (**E'''**).

The online version of this article includes the following figure supplement(s) for figure 3:

**Figure supplement 1.** GATA3 is downregulated but IKZF2 is upregulated in *Rbm24*-deficient hair cells (HCs).

---

*2022*; *Bardhan et al., 2019*; *Sun et al., 2021*), were altered in the absence of RBM24. For analyzing GATA3, mosaic *Rbm24* mutant mice were produced using the CRISPR-stop approach (*Figure 3— figure supplement 1A*). Dual immunostaining of GATA3 and RBM24 revealed that GATA3 was evenly expressed in control OHCs and IHCs at P1 (*Figure 3—figure supplement 1B–B''*). By contrast, the GATA3 level in OHCs lacking RBM24 expression (white arrows in *Figure 3—figure supplement 1C–C''*) was weaker than that in neighboring OHCs that retained RBM24 expression (yellow arrows in *Figure 3—figure supplement 1C–C''*). This result suggests that GATA3 is downregulated in *Rbm24*-deficient OHCs. OHCs degenerate in *Gata3^+/-^* (*Bardhan et al., 2019*), and thus the downregulation of GATA3 might contribute to the cell death of *Rbm24*-deficient OHCs.

For analyzing IKZF2, mosaic *Rbm24* mutant mice were produced using zygotes obtained by breeding male *Ikzf2^V5/+^* mice with WT female mice (*Figure 3—figure supplement 1D*). Notably, the C-terminus of IKZF2 here was fused with three V5 tags to allow IKZF2 protein to be visualized using a V5 antibody (*Li et al., 2023a*; *Bi et al., 2022*). In control *Ikzf2^V5/+^*; *Rbm24^+/+^* mice, V5 (IKZF2) was

evenly distributed in OHCs at P5 (*Figure 3—figure supplement 1E–E″*). By contrast, in the mosaic *Ikzf2*$^{V5/+}$; *Rbm24*$^{-/-}$ mice, the V5 signal in the OHCs that had lost RBM24 expression (yellow arrows in *Figure 3—figure supplement 1F–F″*) was higher than the nearby OHCs expressing RBM24 (white arrows in *Figure 3—figure supplement 1F–F″*). It supports that RBM24 directly or indirectly represses the IKZF2 expression.

## Three *Rbm24* enhancers can drive specific EGFP expression in cochlear HCs

After showing that POU4F3, but not GFI1, is indispensable in mediating *Rbm24* expression, we determined the mechanism by which POU4F3 controls *Rbm24* expression, particularly the role of the CREs of *Rbm24*. According to our previous high-throughput ATAC-seq (transposase-accessible chromatin sequencing) analysis of neonatal cochlear HCs (*Luo et al., 2022*), four CREs of *Rbm24* were identified: one proximal promoter (arrow in *Figure 4A*), and three distal potential enhancers that were defined as Eh1, Eh2, and Eh3 (dotted boxes in *Figure 4A*). Eh1 and Eh2 were located upstream and Eh3 was downstream of the *Rbm24* coding region. Moreover, we reanalyzed the results of POU4F3 Cut&Run assays from one previous study (*Yu et al., 2021*) and found that POU4F3 binds to Eh1, Eh2, and Eh3 but not the *Rbm24* promoter (*Figure 4A*).

Whether Eh1, Eh2, and Eh3 are bona fide *Rbm24* enhancers has remained unknown. We reasoned that if Eh1, Eh2, and Eh3 were *Rbm24* enhancers, one of these, together with the mini-promoter of mouse heat shock protein 68 gene (*Hsp68*) (*Luo et al., 2022*; *Kothary et al., 1989*; *Xu et al., 2021*; *Shu et al., 2022*), would be sufficient to drive specific reporter expression in cochlear HCs, and to test this, we established three transgenic mouse strains: Eh1-EGFP+ (*Figure 4B–C″*), Eh2-EGFP+ (*Figure 4D–E″*), and Eh3-EGFP+ (*Figure 4F–G″*); in these strains, EGFP expression would be driven by the mini-promoter of *Hsp68* and Eh1 or Eh2 or Eh3, respectively. The mini-promoter of *Hsp68* alone is reported to be incapable of driving EGFP expression (*Sun et al., 2022*), and in contrast to this, we detected strong EGFP expression through whole-mount analysis in all three transgenic lines (*Figure 4B, D, and F*). Moreover, dual labeling for EGFP and MYO7A in cryosectioned cochleae showed that EGFP was specifically expressed in IHCs and OHCs in all three strains at P1 (*Figure 4C–C″, E–E″, and G–G″*). Collectively, our transgenic assay results suggest that Eh1, Eh2, and Eh3 are *Rbm24* enhancers, and further that POU4F3 regulates *Rbm24* expression primarily by binding to the *Rbm24* enhancers.

## Forced RBM24 expression fails to alleviate the degeneration of *Pou4f3*$^{-/-}$ HCs

Our results thus far clearly showed that POU4F3 can regulate *Rbm24* expression, but whether ectopic RBM24 expression can alleviate the degeneration of *Pou4f3*$^{-/-}$ HCs remained unknown. Thus, we first established a new conditional mouse model, *Rosa26*$^{loxp-stop-loxp-Rbm24*3\times HA/+}$ (*Rosa26*$^{Rbm24/+}$ in short) (*Figure 5A and B*), and to turn on ectopic RBM24 expression in cochlear HCs, we further crossed the *Rosa26*$^{Rbm24/+}$ strain with the strain *Atoh1*$^{Cre/+}$, which targets a majority of cochlear HCs and a fraction of supporting cells (SCs) (*Yang et al., 2010b*). Notably, in the *Rosa26*$^{Rbm24/+}$ strain, RBM24 was tagged with three HA fragments, which allowed endogenous and ectopic RBM24 to be distinguished. In control *Pou4f3*$^{+/+}$ mice (*Figure 5C–C″*) and germline-stable *Pou4f3*$^{-/-}$ mice (*Figure 5D–D″*), MYO7A+HCs did not express HA (RBM24), whereas almost all MYO7A+HCs expressed HA (RBM24) in *Pou4f3*$^{+/+}$; *Atoh1*$^{Cre/+}$; *Rosa26*$^{Rbm24/+}$ mice at P1 (white arrows in *Figure 5E–E″*). As expected, besides HCs, we detected a few SCs expressing HA (RBM24) but not MYO7A (orange arrows in *Figure 5E–E″*). The cochlear HCs expressing HA (RBM24) appeared normal, suggesting that the cells can tolerate the additional RBM24 expression by P1. Collectively, these results supported the conclusions that the *Rosa26*$^{Rbm24/+}$ model was successfully generated and that ectopic RBM24 was expressed in cochlear HCs in *Pou4f3*$^{+/+}$; *Atoh1*$^{Cre/+}$; *Rosa26*$^{Rbm24/+}$ mice.

In contrast to our expectation, ectopic RBM24 expression failed to mitigate the degeneration of *Pou4f3*$^{-/-}$ HCs. Severe degeneration of HCs (both IHCs and OHCs) occurred in *Pou4f3*$^{-/-}$; *Atoh1*$^{Cre/+}$; *Rosa26*$^{Rbm24/+}$ mice (*Figure 5F–F″*) and was indistinguishable from that in *Pou4f3*$^{-/-}$ mice (*Figure 5D–D″*), except that the surviving HCs expressed RBM24 in the *Pou4f3*$^{-/-}$; *Atoh1*$^{Cre/+}$; *Rosa26*$^{Rbm24/+}$ mice (arrows in *Figure 5F–F″*). This result suggests that restoration of RBM24 expression alone is not sufficient to prevent *Pou4f3*$^{-/-}$ cochlear HCs from undergoing degeneration.

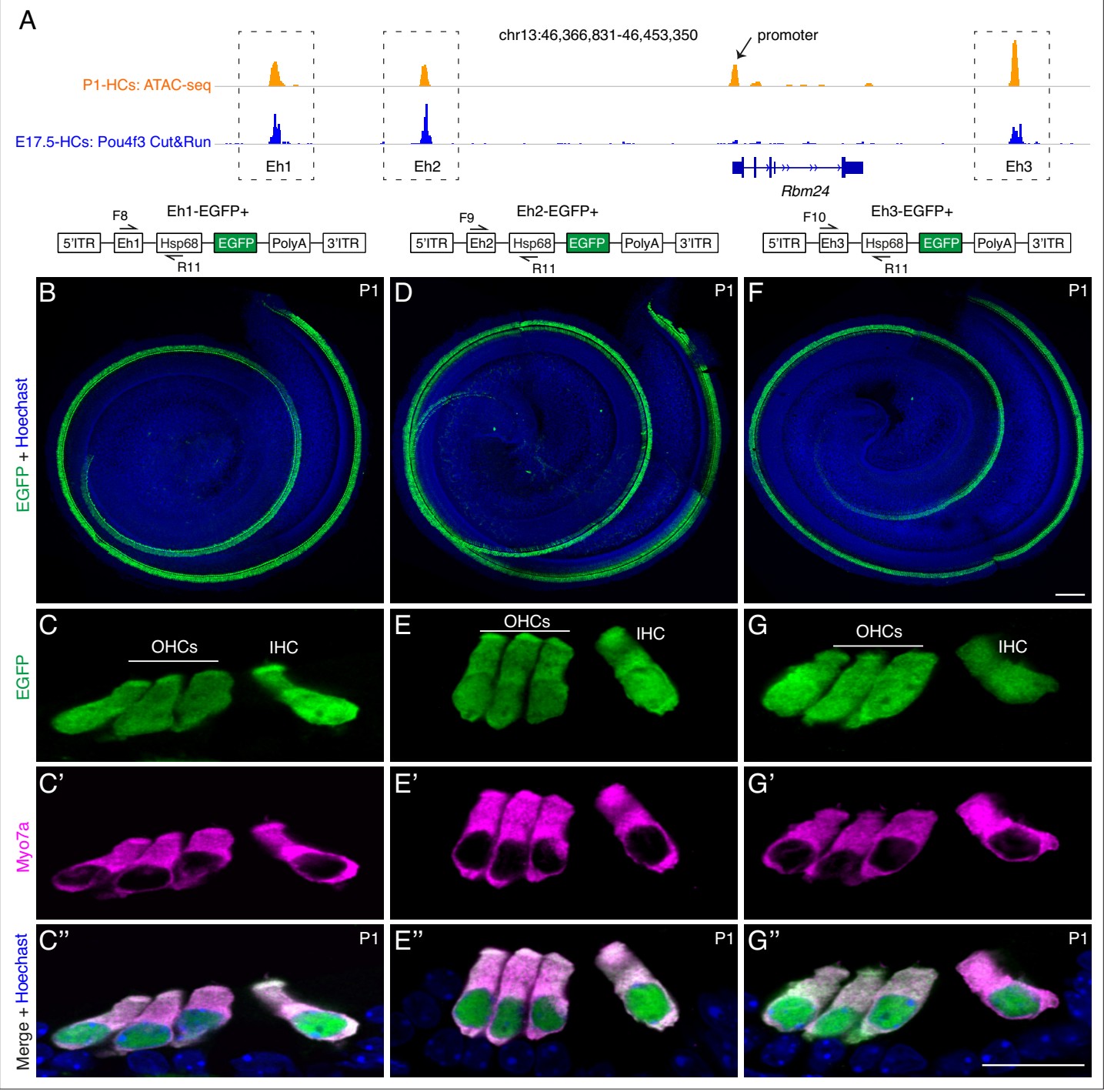

**Figure 4.** Three *Rbm24* enhancers are sufficient to drive specific EGFP expression in cochlear hair cells (HCs). (**A**) Visualization of ATAC-seq (top row, orange) of P1 cochlear HCs (from our previous study, GSE181311) and POU4F3 Cut&Run assay (bottom row, blue) of E17.5 cochlear HCs (from another study, GSE150391) by using Integrative Genomics Viewer. Black arrow: *Rbm24* promoter; dotted boxes: three *Rbm24* enhancers (Eh1, Eh2, and Eh3). (**B–G"**) Dual labeling for EGFP and MYO7A in Eh1-EGFP+ (**B–C"**, n = 7), Eh2-EGFP+ (**D–E"**, n = 6), and Eh3-EGFP+ (**F–G"**, n = 4) mice at P1. Only the EGFP channel (plus nuclear staining) is shown for whole-mount samples (**B, D, F**). In cryosection cochlear samples, EGFP labeling overlapped with HC marker MYO7A. OHCs: outer hair cells; IHC: inner hair cell. Scale bars: 100 µm (**F**), 20 µm (**G"**).

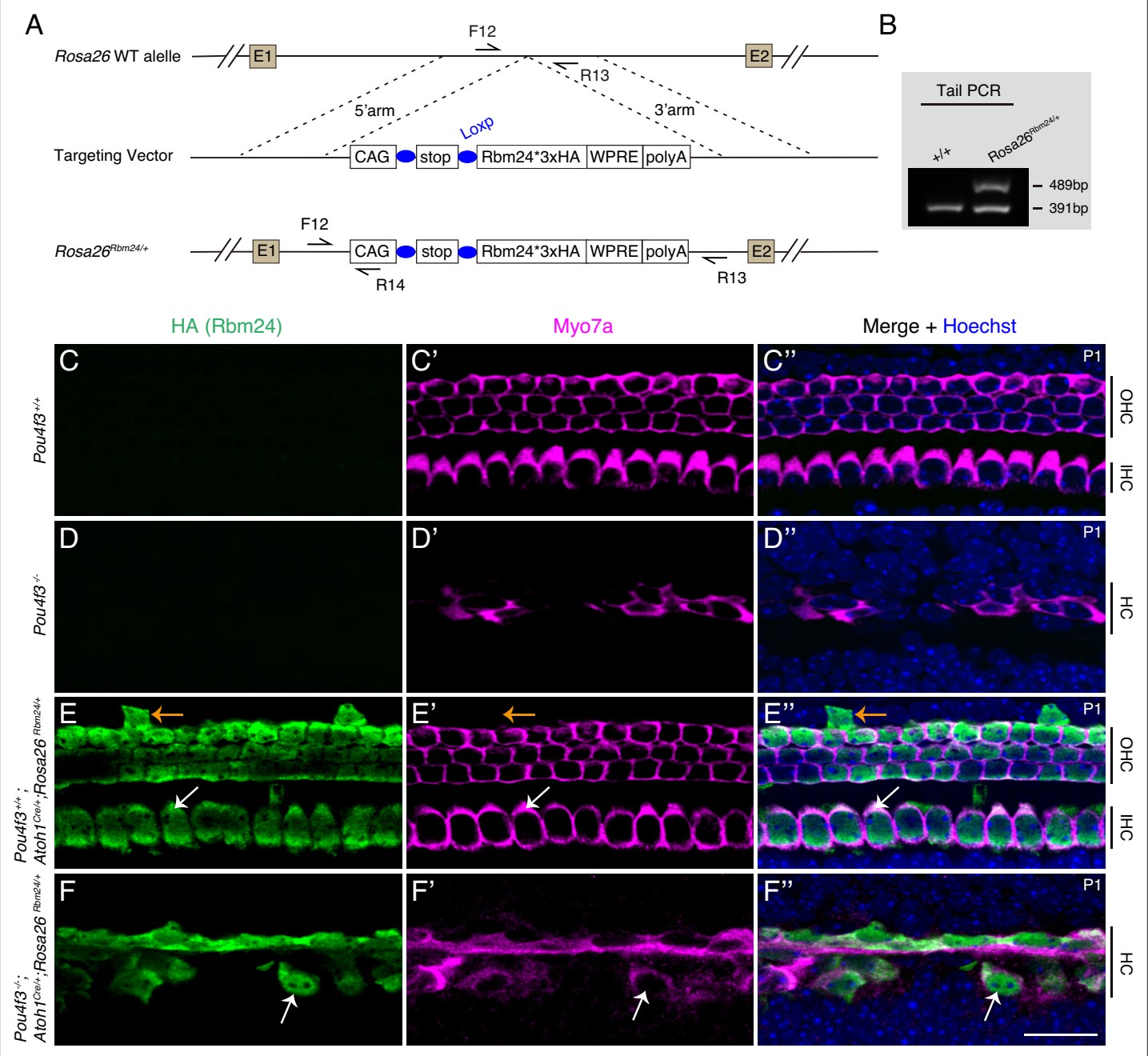

**Figure 5.** Forced expression of RBM24 fails to rescue outer hair cell (OHC) degeneration in *Pou4f3⁻/⁻* mutants. (**A**) Generation of *Rosa26^loxp-stop-loxp-Rbm24*3×HA/+* (*Rosa26^Rbm24/+*) mouse strain. The C-terminus of ectopic RBM24 was fused with three HA tags. (**B**) One example gel image of tail-DNA PCR used to distinguish WT (+/+) and knockin (*Rosa26^Rbm24/+*) mice. (**C–F"**) Dual staining of HA (RBM24) and hair cell (HC) marker MYO7A in mice featuring four distinct genotypes at P1: *Pou4f3^+/+* (**C–C"**, n = 3), *Pou4f3⁻/⁻* (**D–D"**, n = 3), *Pou4f3^+/+; Atoh1^Cre/+; Rosa26^Rbm24/+* (**E–E"**, n = 3), and *Pou4f3⁻/⁻; Atoh1^Cre/+; Rosa26^Rbm24/+* (**F–F"**, n = 3). Ectopic HA (RBM24) was not detected in *Pou4f3^+/+* (**C**) and *Pou4f3⁻/⁻* (**D**) mice, but was detected in both *Pou4f3^+/+; Atoh1^Cre/+; Rosa26^Rbm24/+* (**E**) and *Pou4f3⁻/⁻; Atoh1^Cre/+; Rosa26^Rbm24/+* (**F**) mice. White arrows in (**E–E"**): one MYO7A+ inner hair cell (IHC) that expressed ectopic HA (RBM24). HCs appeared normal by P1, despite the induction of ectopic RBM24 expression. Orange arrows in (**E–E"**): one supporting cell (SC) that expressed HA (RBM24) but not MYO7A. White arrows in (**F–F"**): one remaining MYO7A+ HC that expressed ectopic RBM24. Overall, many of the *Pou4f3⁻/⁻* HCs degenerated even though ectopic RBM24 expression was induced. IHC: inner hair cell. Scale bar: 20 μm (**F"**).

The online version of this article includes the following source data for figure 5:

**Source data 1.** The original agarose gel image in *Figure 5B* (*Rosa26^+/+* and *Rosa26^Rbm24/+*).

**Source data 2.** File containing *Figure 5B* and the original agarose gel analysis with highlighted bands and sample labels.

## Discussion

### Molecular mechanisms underlying cochlear HC survival

Mammalian sound-receptor HCs are vulnerable to various genetic mutations, environmental ototoxic factors, and aging. HC degeneration is one of the primary reasons for human sensorineural hearing impairment (*Petit et al., 2023*), and several genes whose mutations lead to HC degeneration starting at different ages have been identified previously. For example, HC development is severely defective in *Atoh1*-/- mutants (*Bermingham et al., 1999*; *Fritzsch et al., 2005*), and *Atoh1*-/- cochlear sensory cells undergo apoptosis. However, whether the absence of HCs in *Atoh1*-/- mice is because of HCs not being produced or dying immediately after initial emergence remains unclear, partly due to the difficulty of unambiguously defining nascent HCs by using molecular markers.

Unlike in *Atoh1*-/- mutants, in both *Pou4f3*-/- and *Gfi1*-/- mice (*Xiang et al., 1998*; *Wallis et al., 2003*; *Hertzano et al., 2004*), initial cochlear HC development is normal but becomes defective at perinatal ages, consistent with the notion that POU4F3 and GFI1 are dispensable for HC fate specification but necessary for subsequent differentiation and survival. Intriguingly, caspase-3 is active during HC death in *Pou4f3*-/- mutants, and the antiapoptotic factor z-VAD-fmk exerts a protective effect on *Pou4f3*-/- HCs between E14.5 and E16.5 (*Atar and Avraham, 2010*). Moreover, we noted here milder overall HC degeneration in *Gfi1*-/- mice than in *Pou4f3*-/- mice at P1 because only OHC degeneration occurred in *Gfi1*-/- mice but both IHCs and OHCs were degenerated in *Pou4f3*-/- mice (*Figure 1—figure supplement 2*, *Figure 2—figure supplement 3*).

*Gfi1* is a recognized target of POU4F3 (*Hertzano et al., 2004*), but whether forced GFI1 expression can alleviate the degeneration of *Pou4f3*-/- HCs is unknown. Similarly, *Pou4f3* is one of the target genes regulated by ATOH1 (*Yu et al., 2021*), but whether forced expression of POU4F3 or GFI1, or both, can mitigate the developmental defects of *Atoh1*-/- HCs remains undetermined. Future studies must address these questions to enable comprehensive understanding of the mechanisms underlying the maintenance of cochlear HC survival.

### Roles of RBM24 and its regulation during cochlear HC development

Shortly after cochlear HCs emerge, RBM24 expression begins and then is maintained permanently, as revealed by transcriptomic analyses and antibody staining (*Bi et al., 2022*; *Grifone et al., 2018*; *Sun et al., 2021*; *Liu et al., 2022a*; *Ranum et al., 2019*). RBM24 is not necessary in the early phase of cochlear HC development because *Rbm24*-/- HC development is normal by P1, but RBM24 is required for OHC survival after birth (*Wang et al., 2021*); *Rbm24*-/- OHCs, but not IHCs, are degenerated by P19 (*Wang et al., 2021*). RBM24 is also involved in mRNA stability and pre-mRNA alternative splicing of several genes, including *Cdh23* and *Pcdh15*, which are crucial for the development of HC stereocilia (*Wang et al., 2023*; *Zheng et al., 2021*; *Liu et al., 2022b*). Our study also revealed that GATA3 expression is decreased and IKZF2 expression is increased in OHCs in the absence of RBM24, which could be a direct or indirect effect of RBM24 loss. Nonetheless, single-cell transcriptomic analysis has not yet been performed on *Rbm24*-/- IHCs or OHCs, and future investigation is necessary to clarify the detailed molecular mechanism underlying *Rbm24*-/- HC degeneration.

What are the *trans*-acting factors involved in regulating *Rbm24* expression? First, this study has provided strong genetic evidence indicating that POU4F3 is necessary for turning on *Rbm24* expression; in the absence of POU4F3, *Rbm24* expression was not triggered. Moreover, the normal expression of POU4F3 in *Rbm24*-/- HCs confirmed that *Pou4f3* is epistatic to *Rbm24* and that RBM24 is dispensable for *Pou4f3* expression. Besides POU4F3, ATOH1 appears to regulate *Rbm24* expression, as per two lines of evidence: (1) RBM24 expression is lost in *Atoh1*-/- cochlear HCs (*Cai et al., 2015*) and (2) *Rbm24* is one of the ATOH1-binding targets revealed by the ATOH1 Cut&Run assay (*Luo et al., 2022*; *Yu et al., 2021*). Notably, ATOH1 binds to both the *Rbm24* promoter and the three *Rbm24* enhancers (Eh1, Eh2, and Eh3) (*Luo et al., 2022*; *Yu et al., 2021*), whereas POU4F3 only binds to the *Rbm24* enhancers (*Figure 4*). Thus, POU4F3 and ATOH1 likely cooperate to regulate *Rbm24* expression, and either *Pou4f3* or *Atoh1* mutation leads to repression of *Rbm24* expression. Future conditional *Pou4f3* loss-of-function studies are necessary to determine whether *Rbm24* expression in adult HCs requires POU4F3.

### *Gfi1* and *Rbm24* are expressed independently of each other

We initially hypothesized that POU4F3 regulates *Rbm24* expression through GFI1. However, this hypothesis was not supported by our observation that RBM24 expression is normal in *Gfi1*[-/-] HCs (*Figure 2*, *Figure 2—figure supplement 3*). Moreover, in the absence of RBM24, GFI1 expression was not altered in cochlear HCs. Thus, *Rbm24* expression and *Gfi1* expression appear to be independent of each other. This might be due to the functional difference between POU4F3 and GFI1 during cochlear HC development. Although both POU4F3 and GFI1 are necessary for promoting the expression of genes involved in HC differentiation (*Xiang et al., 1997*; *Erkman et al., 1996*; *Hertzano et al., 2004*; *Matern et al., 2020*), GFI1, but not POU4F3, also represses the preceding expression of neural genes in nascent HCs (*Matern et al., 2020*).

Another observation here relevant to *Gfi1* isoforms is noteworthy. It is known that one efficient sgRNA is adequate for inducing homozygous gene inactivation by using CRISPR-stop (*Wang et al., 2021*). Here, we used four *Gfi1* sgRNAs distributed across distinct exons (*Figure 2A*). Interestingly, we successfully established one *Gfi1* mutant by using sgRNA-5 alone, and this, in principle, should effectively pre-stop GFI1 translation in exon 1, which codes for the SNAG repressor domain; the obtained *Gfi1* mutant presented the HC degeneration phenotype, although HA (GFI1) remained detectable in HCs. This agrees with the notion that the *Gfi1* mutant lacking the SNAG domain is equivalent to the *Gfi1*-null model (*Fiolka et al., 2006*). Thus, *Gfi1* is likely expressed as multiple unknown isoforms, many of which might be recognized by the HA-tag antibody because HA is tagged to the last exon (exon 6). However, when we used the four *Gfi1* sgRNAs together, we were able to both reproduce the HC degeneration phenotype and completely block HA (GFI1) expression.

### Why does ectopic RBM24 fail to alleviate the degeneration of *Pou4f3*[-/-] HCs?

Upon confirming the epistatic genetic interaction between *Pou4f3* and *Rbm24*, we predicted that ectopic RBM24 expression should be able to, at least partly, alleviate the degeneration of *Pou4f3*[-/-] HCs, similar to what we recently reported in our study on INSM1 and IKZF2 in cochlear OHCs (*Li et al., 2023a*). However, RBM24 failed to rescue the HC degeneration in *Pou4f3*[-/-] mice. Two potential explanations for this finding are the following: (1) although *Rbm24* expression is modulated by POU4F3, RBM24 is not directly involved in the pathway regulating HC survival; accordingly, cell death occurs later in *Rbm24*[-/-] HCs than in *Pou4f3*[-/-] HCs. (2) *Rbm24* is a key POU4F3 downstream target, but forced expression of RBM24 alone cannot compensate for the loss of POU4F3 or other known POU4F3 targets such as orphan thyroid nuclear receptor *Nr2f2* and *Caprin-1* (*Tornari et al., 2014*; *Towers et al., 2011*); CAPRIN-1 is recruited to stress granules in cochlear HCs exposed to ototoxic trauma (*Towers et al., 2011*).

### Potential application of the three *Rbm24* enhancers in cochlear HC gene therapy

Gene therapy is a promising strategy for restoring hearing capacity in humans with inherited gene mutations causing hearing impairment (*Petit et al., 2023*), and a few such therapy examples have been reported to date, including *Otoferlin* and *vGlut3* gene-replacement therapies (*Akil et al., 2019*; *Tang et al., 2023*; *Al-Moyed et al., 2019*; *Akil et al., 2012*). Currently, therapeutic cDNAs are primarily delivered into HCs by using an adeno-associated virus (AAV) vector. Although several AAVs have been reported (*Tan et al., 2019*; *Wu et al., 2021*), the vectors transfect cochlear cells non-selectively and HC-specific AAVs are not yet available. A solution to this problem is the following: instead of the CAG/CMV ubiquitous promoter widely used in current AAVs (*Isgrig et al., 2019*; *Landegger et al., 2017*), using any one of the three *Rbm24* enhancers together with the *Hsp68* mini-promoter (in brief *Rbm24-Hsp68*) should generate an AAV that would allow HC-specific transfection. Moreover, *Rbm24* is permanently expressed in cochlear HCs, and if *Rbm24-Hsp68* AAV acts as expected, it

would represent a powerful tool for future HC-specific gene therapy at all postnatal ages, with potential applications in treating clinical human deafness.

# Materials and methods

**Key resources table**

| Reagent type (species) or resource | Designation | Source or reference | Identifiers | Additional information |
|---|---|---|---|---|
| Strain, strain background (*Mus musculus*) | *Atoh1*[*Cre/+*] | *Yang et al., 2010b* | Lin Gan lab (Augusta University) | A knock-in mouse strain |
| Strain, strain background (*M. musculus*) | *Rosa26*[*-loxp-stop-loxp-tdTomato*] (*Ai9*) | Stock# 007905 | The Jackson Laboratory | A knock-in mouse strain |
| Strain, strain background (*M. musculus*) | *Ikzf2*[*V5/+*] | Stock# 038463 | The Jackson Laboratory | A knock-in mouse strain |
| Strain, strain background (*Mus musculus*) | *Pou4f3* mutant | This paper | Available upon request from Liu Lab | Constructed by CRISPR-stop |
| Strain, strain background (*M. musculus*) | *Gfi1* mutant | This paper | Available upon request from Liu Lab | Constructed by CRISPR-stop |
| Strain, strain background (*M. musculus*) | *Rbm24* mutant | This paper | Available upon request from Liu Lab | Constructed by CRISPR-stop |
| Strain, strain background (*M. musculus*) | *Pou4f3* null | This paper | Available upon request from Liu Lab | Constructed by CRISPR /Cas9 |
| Strain, strain background (*M. musculus*) | *Gfi1* null | This paper | Available upon request from Liu Lab | Constructed by CRISPR /Cas9 |
| Strain, strain background (*M. musculus*) | *Gfi1*[*3xHA-P2A-Cre*] | This paper | Available upon request from Liu Lab | A knock-in mouse strain |
| Strain, strain background (*M. musculus*) | Eh1-EGFP+ | This paper | Available upon request from Liu Lab | A transgenic mouse line |
| Strain, strain background (*M. musculus*) | Eh2-EGFP+ | This paper | Available upon request from Liu Lab | A transgenic mouse line |
| Strain, strain background (*M. musculus*) | Eh3-EGFP+ | This paper | Available upon request from Liu Lab | A transgenic mouse line |
| Strain, strain background (*M. musculus*) | Rosa26[*CAG-Isl-Rbm24*3xHA/+*] | This paper | Available upon request from Liu Lab | A knock-in mouse strain for ectopic Rbm24 expression |
| Antibody | Anti-RBM24 (rabbit polyclonal) | Proteintech | 18178-1-AP | IF (1:500) |
| Antibody | Anti-POU4F3 (mouse monoclonal) | Santa Cruz | sc-81980 | IF (1:500) |
| Antibody | Anti-POU4F3 (rabbit polyclonal) | Novus Biologicals | NBP1-88349 | IF (1:500) |
| Antibody | Anti-HA (rat monoclonal) | Roche | 11867423001 | IF (1:500) |
| Antibody | Anti-MYO7A (rabbit polyclonal) | Proteus Biosciences | 25-6790 | IF (1:500) |
| Antibody | Anti-MYO7A (mouse monoclonal) | DSHB | MYO7A 138-1 | IF (1:500) |
| Antibody | Anti-CTIP2 (rat monoclonal) | Abcam | ab18465 | IF (1:500) |
| Antibody | Anti-INSM1 (guinea pig) | *Jia et al., 2015* | A kind gift from Dr. Carmen Birchmeier | IF (1:6000) |
| Antibody | Anti-GFP (chicken polyclonal) | Abcam | ab13970 | IF (1:500) |

*Continued on next page*

*Continued*

| Reagent type (species) or resource | Designation | Source or reference | Identifiers | Additional information |
|---|---|---|---|---|
| Antibody | Anti-GATA3 (goat polyclonal) | R&D Systems | AF2605 | IF (1:500) |
| Antibody | Anti-V5 (mouse monoclonal) | Bio-Rad | MCA1360 | IF (1:500) |

## Mice

The *Atoh1*$^{Cre/+}$ model was kindly provided by Dr. Lin Gan (Augusta University, USA). The *Rosa26-loxp-stop-loxp-tdTomato* (Ai9)/+ strain (Jax#: 007905) was from The Jackson Laboratory. The *Ikzf2*$^{V5/+}$ mouse strain is described in detail in our previous reports (*Li et al., 2023a*; *Bi et al., 2022*). All mice were bred and raised in an SPF-level animal room, and all animal procedures were performed according to the guidelines (NA-032-2022) of the Institutional Animal Care and Use Committee (IACUC) of the Institute of Neuroscience (ION), Center for Excellence in Brain Science and Intelligence Technology, Chinese Academy of Sciences.

## One-step generation of homozygous *Pou4f3*, *Gfi1*, or *Rbm24* mutants by using CRISPR-stop

The detailed protocol for using CRISPR-stop to generate homozygous gene mutants is described in our previous reports (*Wang et al., 2021*; *Zhang et al., 2018*). Briefly, efficient pre-tested sgRNAs (*Supplementary file 1*) and hA3ABE3 were co-injected into one-cell-stage mouse zygotes that were then transplanted into pseudopregnant female mice, which gave birth to F0 mice. The F0 mice carrying the expected homozygous mutation (pre-emergence of protein translation stop codon) were identified using Sanger sequencing of tail-DNA PCR samples and were immediately ready for analysis.

## Generating *Pou4f3*- or *Gfi1*-null mutants harboring large DNA fragment deletions by using CRISPR/Cas9

To construct germline-stable null mutants of either *Pou4f3* or *Gfi1*, *Cas9* mRNA, two efficient pre-tested sgRNAs located at the proximal and distal ends of the targeted gene, and a single-stranded DNA donor (120 bp) (*Supplementary file 2*) were co-injected into one-cell-stage WT zygotes. Notably, the left and right halves (60 bp each) of the single-stranded DNA donor were homologous to the 5′ and 3′ ends of the targeted gene, respectively. The post-injected zygotes were transplanted into pseudopregnant females, which gave birth to F0 mice; the F0 mice were subject to tail-DNA PCR screening with the primers listed in *Supplementary file 2*, and the mice harboring the designed large DNA deletion between the two sgRNAs were identified and further bred with WT mice to establish the germline-stable mutants (F1 or afterward).

## Construction of *Gfi1*$^{3◊HA-P2A-Cre}$ knockin mouse strain

An sgRNA against *Gfi1* (5′-ATGGACTCAAATGAGTACCC-3′), *Cas9* mRNA, and the targeting vector (*Figure 2—figure supplement 1B*) were co-injected into one-cell-stage WT mouse zygotes. The targeting vector comprised three portions: the 5′ homologous arm (800 bp), the 3′ homologous arm (800 bp), and the region between the 5′ and 3′ arms that contained three HA fragments followed by 2A-Cre. The F0 mice with the potential gene targeting (*Figure 2—figure supplement 1C*) were screened using tail-DNA PCR and then crossed with WT mice to produce F1 mice; these F1 mice were confirmed using tail-DNA PCR again and further screened using Southern blotting. The detailed Southern blotting protocol is described in our previous report (*Li et al., 2018*). Tail-DNA PCR was used for routine genotyping, and the knockin (*Gfi1*$^{HA-Cre/+}$) and WT alleles were distinguished using the primers F4, R5, and R6 (*Supplementary file 3*).

## Construction of Eh1-EGFP+, Eh2-EGFP+, and Eh3-EGFP+ transgenic reporter lines

The three transgenic reporter mouse lines used here, Eh1-EGFP+, Eh2-EGFP+, and Eh3-EGFP+, were produced using the same procedures. The core DNA sequences of each enhancer (Eh1, Eh2, or Eh3

in *Figure 4A*), the mRNA encoding PiggyBac transposase, and the PiggyBac vector (*Figure 4B, D, and F*) were co-injected into one-cell-stage WT mouse zygotes. The PiggyBac vector contained the Eh1/Eh2/Eh3 core DNA sequence (*Supplementary file 2*), the mini-promoter of mouse *Hsp68*, and the EGFP coding sequence, and the vector was randomly integrated into the mouse genome by the transposase. The F0 mice harboring the PiggyBac vector were screened using tail-DNA PCR and further bred with WT mice to produce germline-stable F1 transgenic reporter strains. The samples analyzed in *Figure 4B–G"* were from F1 or later generations. The primers used for genotyping transgenic reporter strains were F8, F9, F10, and the common primer R11 (primer sequences are listed in *Supplementary file 3*).

## Generation of *Rosa26$^{CAG-lsl-Rbm24*3×HA/+}$* mouse model

The *Rosa26$^{CAG-lsl-Rbm24*3×HA/+}$* (*Rosa26$^{Rbm24/+}$*) knockin mouse strain was constructed through homologous recombination mediated by CRISPR/Cas9. A pre-tested *Rosa26* sgRNA (5'-*ACTCCAGTCTTTCTAG AAGA*-3'), the targeting vector (*Figure 5A*), and *Cas9* mRNA were co-injected into one-cell-stage WT mouse zygotes, and similarly as in other cases, F0 mice with potentially correct gene targeting were screened using tail-DNA PCR and further bred with WT mice to establish germline-stable F1 mice. The WT and knockin (*Rosa26$^{Rbm24/+}$*) alleles were distinguished using the primers F12, R13, and R14 (*Supplementary file 3*).

## Sample processing and immunofluorescence assay

Inner ears were dissected out and fixed in 4% paraformaldehyde (PFA) in PBS (E607016-0500, Sangon Biotech) at 4°C overnight. For obtaining cochlear cryosections, inner ears were dehydrated in 30% sucrose (V900116, Sigma) at 4°C before embedding into optimal cutting temperature (OCT) compound (4583, SAKURA), and then slices were cut at 14 µm thickness. The detailed protocol for immunofluorescence staining is described in our previous report (*Liu et al., 2010*). The following primary antibodies were used: anti-RBM24 (rabbit, 1:500, 18178-1-AP, Proteintech), anti-POU4F3 (mouse, 1:500, sc-81980, Santa Cruz), anti-POU4F3 (rabbit, 1:500, NBP1-88349, Novus Biologicals), anti-HA (rat, 1:500, 11867423001, Roche), anti-MYO7A (rabbit, 1:500, 25-6790, Proteus Biosciences), anti-MYO7A (mouse, 1:500, MYO7A 138-1, Developmental Studies Hybridoma Bank), anti-CTIP2 (BCL11B) (rat, 1:500, ab18465, Abcam), anti-INSM1 (guinea pig, 1:6000, a kind gift from Dr. Carmen Birchmeier from Max Delbrueck Center for Molecular Medicine, Germany), anti-GFP (chicken, 1:500, ab13970, Abcam), anti-GATA3 (goat, 1:500, AF2605, R&D Systems), and anti-V5 (mouse, 1:500, MCA1360, Bio-Rad). Various corresponding Alexa Fluor-conjugated secondary antibodies were used for detecting the primary antibodies, and Hoechst 33342 (1:1000, H3570, Thermo Fisher Scientific) was used for nuclear DNA staining.

After staining, the whole-mount or cryosection samples were mounted with Prolong Gold antifade medium (P36930, Thermo Fisher Scientific) at room temperature for 12 hr. Samples were scanned using a Nikon C2 or Nikon NiE-A1 Plus confocal microscope, and ImageJ software was used to process the confocal images.

## Cell quantification and statistical analysis

Before immunofluorescence staining, each cochlear sample was grossly separated into three portions of distinct lengths, and each portion of the same cochlea was initially scanned using a confocal microscope at low magnification (×10 lens). After calculating the total length of each cochlea, the cochlear sample was precisely divided into basal, middle, and apical turns of equal length. Subsequently, for cell counting in experiments (*Figure 1—figure supplement 2D*), an ~200 µm stretch of the sensory epithelium in each turn was scanned using a confocal microscope at high magnification (×60 lens) and the number of HCs was determined. In *Pou4f3$^{-/-}$* cochleae, the percentage of surviving HCs was calculated by normalizing the number of remaining HCs against their counterparts in WT mice. All cell numbers are presented as means ± SD. For statistical analyses, we used GraphPad Prism 6.0 software and performed Student's *t*-tests with Bonferroni correction.

## Institutional review board statement

The animal study protocol (NA-032-2022) was approved by the Institutional Animal Care and Use Committee (IACUC) of the Institute of Neuroscience (ION), Center for Excellence in Brain Science and Intelligence Technology, Chinese Academy of Sciences.

## Data availability statement

All mouse strains or other reagents reported in this study are available upon reasonable request to the corresponding author.

## Acknowledgements

We thank Dr. Qian Hu of the Optical Imaging Facility of the Institute of Neuroscience (ION) for support with image analysis; and Ms. Qian Liu from the Department of Embryology of ION animal center for helping us in transplanting zygotes into pseudopregnant female mice. We thank Dr. Lin Gan (Augusta University, USA) for the Atoh1$^{Cre/+}$ strain.

## Additional information

### Funding

| Funder | Grant reference number | Author |
|---|---|---|
| Ministry of Science and Technology of the People's Republic of China | 2021YFA1101804 | Zhiyong Liu |
| National Natural Science Foundation of China | 32321163648 | Zhiyong Liu |

The funders had no role in study design, data collection and interpretation, or the decision to submit the work for publication.

### Author contributions

Guangqin Wang, Conceptualization, Formal analysis, Investigation, Methodology, Writing - original draft; Yunpeng Gu, Formal analysis, Investigation; Zhiyong Liu, Conceptualization, Supervision, Funding acquisition, Investigation, Writing - original draft, Project administration, Writing - review and editing

### Author ORCIDs

Guangqin Wang ⦿ http://orcid.org/0000-0002-8613-7619
Zhiyong Liu ⦿ http://orcid.org/0000-0002-9675-1233

### Ethics

All mice were bred and raised in an SPF-level animal room and all animal procedures were performed according to the guidelines (NA-032-2022) of the Institutional Animal Care and Use Committee (IACUC) of the Institute of Neuroscience (ION), Center for Excellence in Brain Science and Intelligence Technology, Chinese Academy of Sciences.

Reviewer #1 (Public Review): https://doi.org/10.7554/eLife.90025.3.sa1
Reviewer #2 (Public Review): https://doi.org/10.7554/eLife.90025.3.sa2
Author Response https://doi.org/10.7554/eLife.90025.3.sa3

## Additional files

### Supplementary files
• Supplementary file 1. List of sgRNA sequences.
• Supplementary file 2. DNA sequences of ssDNA donor and Rbm24 enhancers.

- Supplementary file 3. DNA sequences of genotyping primers.
- MDAR checklist

## Data availability

All data generated or analysed during this study are included in the manuscript and supporting files; source data files have been provided for *Figures 1, 2 and 5*, *Figure 1—figure supplement 2*, *Figure 2—figure supplements 1 and 3*.

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
