## [Editor Report · eLife assessment]

In this **valuable** study, the authors explore regulatory cascades governing mammalian cochlear hair cell development and survival. They confirm previous studies that the transcription factors Pou4f3 and Gfi1 are necessary for hair cell survival and use **compelling** evidence to demonstrate that the RNA-binding protein gene RBM24 is regulated by Pou4f3, but not Gfi1. These findings will be of interest to those working on hearing loss and hold significance for viral gene delivery methods aiming to manipulate gene expression.

---

## [Referee Report · Reviewer #1 (Public Review)]

Wang and colleagues recently demonstrated the essential role of RBM24 (RNA-binding motif protein 24a) in the development of mouse hair cells (source: https://doi.org/10.1002/jcp.31003). In this study, they further expand on their findings by revealing that Rbm24 expression is absent in Pou4f3 mutant mice but not in Gfi1 mutant mice. This observation suggests that POU4F3 acts as an upstream regulator of Rbm24. The researchers effectively demonstrate that POU4F3 can bind to and regulate Rbm24 through three distant enhancers, which are located in open chromatin regions and are bound by POU4F3. Lastly, Wang and colleagues discovered that ectopic expression of Rbm24 was unable to prevent the degeneration of POU4F3 null hair cells.

The findings in this manuscript hold great significance as they provide additional insights into the transcriptional cascades crucial for hair cell development. The discovery of enhancers capable of driving transgene expression specifically in hair cells holds promising therapeutic implications. The figures presented in the study are of excellent quality, the employed techniques are state-of-the-art, the data are accurately represented without exaggeration, and the study demonstrates a high level of rigor.

---

## [Referee Report · Reviewer #2 (Public Review)]

Previous studies have shown that two hair cell transcription factors, Pou4f3 and Gfi1 are both necessary for the survival of cochlear hair cells, and that Gfi1 is regulated by Pou4f3. The authors have previously also shown that mosaic inactivation of the RNA-binding protein RBM24 leads to outer hair cell death.

In the present study, the authors show that hair cells dies in Pou4f3 and Gfi1 mutant mice. They show that Gfi1 is regulated by Pou4f3. Both these observations have been published before. They then show that RBM24 is absent in Pou4f3 knockouts, but not Gfi1 knockouts. They ectopically activate RMB24 in the hair cells of Poui4f3 knockouts, but this does not rescue the hair cell death. Finally the authors validate three RMB24 enhancers that are active in young hair cells and which have been previously shown to bind Pou4f3.

The experiments are well-executed and the data are clear. The results support the conclusions of the paper. The authors have revised the paper slightly, mostly to modify the red/green staining in the figures, and to perform additional analyses of the RBM24 and Ikzf2 mutants, now shown in Supplementary Figure 3.

Much of the work in the paper has been reported before. The result that hair cell transcription factors operate in a network, with some transcription factors activating only a subset of hair cell genes, is an expected result. Since RBM24 is only one of many genes regulated directly by Pou4f3, it is not surprising that it cannot rescue the Pou4f3 knockout hair cell degeneration, and indeed the rationale for attempting such a rescue experiment is not provided by the authors.

The identification of new hair cell enhancers may be of use to investigators wishing to express genes in hair cells.

In sum, this work, although carefully performed, does not shed significant new light on our understanding of hair cell development or survival.

---

## [Author Response]

The following is the authors’ response to the original reviews.

**Reviewer #1 (Recommendations For the Authors):**
(1) While not absolutely necessary - it would be nice to see at least at the in-situ level what happens to the handful of other HC-important transcription factors in the Rbm24 KO (IKZF2, Barlh1, RFX) as the authors did look at Insm1.

Reply: Thanks for your suggested experiments. We agree that knowing whether the genes that are known to be involved in cell survival regulation are changed will provide insights into the mechanisms underlying cell death of Rbm24-/- HCs. Our data showed that Ikzf2 seemed to be upregulated when in the Rbm24-/- HCs, relative to Rbm24+/+ HCs at P5. We also tested Barlh1 and RFX, but we did not obtain confident data to present. Nonetheless, following the reviewer’s logic, we further tested Gata3, another gene involved in HC survival, and found that Gata3 was down-regulated in Rbm24 -/- HCs, compared to Rbm24+/+ HCs. Please refer to the text on lines 12-22 on page 12 and lines 1-10 on page 13, and Figure 3-figure supplement 1.

(2) Major comments: The nomenclature for mouse gene vs. mouse protein needs to be addressed throughout the manuscript. The nomenclature when referring to a mouse gene: gene symbols are italicized, with only the first letter in upper-case (e.g. Rbm24).

The nomenclature when referring to a mouse protein: Protein symbols are not italicized, and all letters are in upper-case (e.g. RBM24).

Reply: Thanks for pointing it out. In the entire manuscript, we have followed the reviewer’s comments to list gene and protein.

(3) Supplemental Figure 2D: Individual data points should be displayed on the bar graph via dots. SEM is not appropriate for this graph as SEM precision with only 3 samples is low. Furthermore, readers are more interested in knowing the variability within samples and not proximity of mean to the population mean, therefore standard deviation (SD) should be used instead.

Reply: We have edited the Figure 1-figure supplement 2D, as suggested. The Figure 1figure supplement 2 legend was updated, too. Please refer to line 21-22 on page 32.

(4) Red/Green should be avoided, especially when both are on the same image (merged immunofluorescence images that are found throughout the manuscript). I highly recommend changing to a color-blind friendly color scheme (such as cyan/green/magenta, cyan/magenta/yellow, etc.) for inclusivity.

Reply: Thanks for pointing it out. We have changed the red to magenta in all our Figures and figure supplements.

(5) Minor comments: As CRISPR-stop is a major method used throughout the paper, a brief explanation is needed for readers to understand what this methodology entails and why it was used. Something along the lines of," The CRISPR-stop technique allows for the introduction of early stop codons without the induction of DNA damage via Cas9 which can cause deleterious effects".

Reply: We have further elaborated how CRISPR-stop works and its advantages. Please refer to lines 8-13 on page 5.

(6) Page 5; line 5 - "Phenotypes occur earlier..." Grammar

Reply: The grammar error was corrected. Please refer to line 4, page 5.

(7) Page 5; line 5 - "Given Pou4f3 is the upstream regulator..." Not proven, rephrase

Reply: We have rephrased this sentence. Please refer to lines 5-6 on page 5.

(8) Supplemental 1A: Fine, Proof of knockout, I wouldn't mention INSM1 being "irregular"

Reply: We have rephrased this sentence. Please refer to lines 2-3 on page 6.

(9) Page 5; line21 - "Alignment of Insm1+ OHCs was not as regular..." Not a good description

Reply: We have rephrased this sentence. Please refer to lines 2-3 on page 6.

(10) Page 6; line11 - "Rbm24 was completely absent.." Redundancy with line 9

Reply: Thanks for pointing it out, and we have removed the redundant sentence.

(11) Page 7 - HA tag should be indicated originally as: Hemaglutinin (HA)

Reply: We have switched “HA” to “Hemaglutinin (HA)”. Please refer to line 15, page 7.

(12) Page 9, line 11- "Determine if autonomous/noncell autonomous." Disagree, cells still clustered in supplemental fig 4.

Reply: We have removed this sentence.

**Reviewer #2 (Recommendations For The Authors):**
The writing of the manuscript is adequate, but it would certainly be improved by professional editing.

Reply: Thanks for the reviewer’s encouraging comments. The revised version of our manuscript has been edited by an English native speaker.